# Leveraging Whole-Genome Resequencing to Uncover Genetic Diversity and Promote Conservation Strategies for Ruminants in Asia

**DOI:** 10.3390/ani15060831

**Published:** 2025-03-13

**Authors:** Qinqian Wang, Ying Lu, Mengfei Li, Zhendong Gao, Dongfang Li, Yuyang Gao, Weidong Deng, Jiao Wu

**Affiliations:** Yunnan Provincial Key Laboratory of Animal Nutrition and Feed, Faculty of Animal Science and Technology, Yunnan Agricultural University, Kunming 650201, Chinayinglu_1998@163.com (Y.L.); mfli_2000@163.com (M.L.); zander_gao@163.com (Z.G.); dfli0927@163.com (D.L.); gaoyy5210@163.com (Y.G.)

**Keywords:** ruminants, whole-genome resequencing, economic traits, genetic structure, genetic diversity

## Abstract

Whole-genome resequencing is a powerful genomic technology for high-resolution identification of genetic variations. In ruminants, it has opened new research frontiers. It is widely used to study genetic diversity, helping understand evolutionary history, population structure, and relationships, which is crucial for conserving endangered breeds and genetic resources. For production performance, it has facilitated the discovery of genes related to milk yield, meat quality, and feed efficiency. Breeders can use this info to select superior animals for more efficient and sustainable farming. The progress in this technology holds great promise for advancing understanding of ruminant biology and improving farming productivity and sustainability.

## 1. Introduction

Ruminants are closely intertwined with human daily life. According to a report by the Food and Agriculture Organization of the United Nations (FAO, https://www.fao.org/faostat/en/#data/FBS (accessed on 24 February 2025), China’s beef production reached 6.78885 million tons in 2023, while mutton production (including goat and sheep meat) totaled 5.31402 million tons. The gross production value of raw milk (from cattle, buffalo, goats, and sheep) amounted to 20,313,291 thousand international dollars (I$), and wool production reached 1,415,246 thousand I$. These figures underscore the pivotal role of ruminants in the global food and related industries. With advancing technology, significant breakthroughs have been made in ruminant genome research.

On 23 April 2009, the USDA Agricultural Research Service and Baylor College of Medicine, leading a consortium of over 300 scientists from 25 countries, announced the completion of cattle genome sequencing using a Hereford cow as the reference sample, employing BAC clone sequencing and whole-genome shotgun sequencing strategies [1]. In 2012, a research team led by Professor Jianquan Liu sequenced the genome of a female yak using whole-genome shotgun sequencing and the Illumina HiSeq 2000 platform, producing a 2657 Mb draft genome [2]. In 2017, a team from Northwest A&F University sequenced the genome of a Marco Polo sheep using whole-genome shotgun sequencing and the Illumina HiSeq 2000 platform, generating 1022.43 Gb of raw reads [3]. In 2019, researchers from the University of Adelaide constructed a high-resolution genome map of a female Mediterranean buffalo using single-molecule sequencing and chromatin conformation capture [4]. In 2024, Professor Menghua Li’s team at China Agricultural University assembled the first complete goat genome (T2T-goat1.0), spanning 2.86 Gb and including 20.96 Mb of the Y chromosome, using PacBio HiFi, Ultra-long ONT, Bionano, and Hi-C technologies [5]. With advancements in sequencing technologies, reduced costs, and improved assembly methods, high-quality genome sequences of an increasing number of species have been published.

With the continuous advancement of whole-genome sequencing (WGS) technologies, researchers are now able to obtain high-quality and complete reference genomes, which has laid a solid foundation for whole-genome resequencing (WGRS) studies in ruminants. By aligning WGRS data to reference genome sequences, this technology enables the detection of various genetic variants [6], identification of candidate genes associated with phenotypic traits [7], and analysis of population genetic diversity and structure [8,9]. Through resequencing analysis, researchers can deeply explore the genetic diversity of ruminants, precisely locate functional genes linked to desirable traits, and significantly accelerate the breeding process of new varieties. This approach plays a pivotal role in the development, utilization, and conservation of ruminant genetic resources.

This review summarizes recent advances in WGRS studies of ruminants and explores its applications in genetic resource conservation. The aim is to provide a scientific foundation for a deeper understanding of the genetic characteristics and diversity of ruminants, thereby promoting their conservation and sustainable development.

## 2. Overview of Whole-Genome Resequencing Technology

### 2.1. Definition

WGRS is a genomic sequencing technology designed for species with known genome sequences, enabling genomic analyses of individuals or populations to identify genetic differences. The core of this technology lies in the application of high-throughput sequencing to perform deep sequencing of genomes. Currently, the most widely used sequencing technologies are next-generation sequencing (NGS) and third-generation sequencing (TGS).

TGS is capable of generating long sequencing reads, typically based on single-molecule sequencing and sequencing in real-time principles, which directly read DNA or RNA molecules, thereby avoiding biases introduced by PCR amplification and improving detection accuracy [10]. This technology excels in identifying structural variations [11] and enables complete genome assembly [12]. In contrast, NGS is renowned for its high-throughput capabilities, featuring mature alignment and assembly algorithms, straightforward data quality control, and high cost-effectiveness [13]. In practical applications, TGS is more suitable for research objectives involving the resolution of complex genomic structures, investigation of gene integrity, detection of structural variations, or exploration of epigenetic modifications, leveraging its long-read advantages. On the other hand, NGS, with its short-read characteristics, is more advantageous for large-scale detection of single-nucleotide polymorphisms (SNPs) and insertions/deletions (InDels), screening of common genetic variants, and large-scale genomic sequencing projects where cost-effectiveness is a priority.

WGRS technology can detect a wide range of genetic variations, including SNPs, structural variations (SVs), copy number variations (CNVs), and InDels [14]. Depending on the research objective, WGRS can be categorized into individual-based and population-based approaches. Individual-based methods include high-coverage haplotype-unresolved whole-genome resequencing (huWGR) and high-coverage haplotype-resolved whole-genome resequencing (hrWGR). Population-based methods include pooled sample sequencing (Pool-seq) and low-coverage individual sequencing (lcWGR) [15].

WGRS has a wide range of applications, including the study of evolutionary processes [16,17], identification of functional genes, as well as its roles in animal and plant breeding [18,19], genetic map construction [20] and disease genomics research [21,22]. With the decreasing cost of sequencing and the development of bioinformatics tools, WGRS has become an efficient and cost-effective method for genomic analysis [23]. Its advantages lie in the comprehensive and precise acquisition of genomic information with high data output, enabling the discovery of numerous key genes associated with traits. As a result, WGRS has become an essential tool in modern biological research, driving advancements in multiple scientific fields.

### 2.2. Technical Advantages

WGRS, single-nucleotide polymorphism (SNP) arrays and WGS are pivotal technologies in genomics research, each offering distinct advantages and limitations in terms of resolution, cost-effectiveness, and applicability for studying genetic variations.

In recent years, WGRS has been widely applied in genomic research due to its distinct advantages, including high resolution, high coverage, cost-effectiveness, and low input requirements. The high resolution of WGRS stems from its ability to perform deep sequencing [24,25], advanced sequencing technologies [26], precise detection of genetic variations [27,28], and rigorous bioinformatics workflows [29]. These features make WGRS a powerful tool for studying genomic complexity and genetic diversity. Its high coverage capability, exemplified by single-molecule real-time sequencing (SMRT), significantly improves the assembly of high-GC-content genomes due to its unbiased sequencing and long-read lengths, enhancing coverage in GC-rich and repetitive regions [30]. Regarding cost-effectiveness, the continuous development of new sequencing platforms has led to a substantial reduction in sequencing costs while significantly increasing data output [31]. Furthermore, WGRS requires minimal input DNA, with starting amounts as low as picograms (pg) [32]. This low-input requirement is particularly advantageous for rare or clinical samples, allowing researchers to perform comprehensive genomic analyses with limited material.

SNP arrays are a crucial tool in genomics research, serving as a high-throughput genotyping technology designed to detect single-nucleotide polymorphisms (SNPs) at specific genomic loci [33]. By utilizing pre-designed probes, SNP arrays can simultaneously analyze thousands to millions of known SNP sites, enabling rapid and efficient genotyping of samples [34]. However, SNP arrays are limited to detecting only the pre-designed SNP loci on the chip, with a finite number of detectable sites [35]. They cannot identify SNPs outside the array or other types of genetic variations. In contrast, whole-genome resequencing (WGRS) sequences the entire genome, providing a comprehensive view of all genetic variations, including SNPs, InDels, CNVs, and SV. SNP arrays typically have a lower cost per sample, particularly when analyzing large sample sizes. They are a cost-effective choice when the research focus is on genotyping known SNPs and does not require whole-genome analysis [36]. On the other hand, WGRS involves extensive sequencing and data analysis, resulting in a higher cost per sample. However, with recent advancements in sequencing technologies, costs have significantly decreased. Moreover, for studies requiring the simultaneous analysis of multiple types of genetic variations, WGRS may prove more cost-effective in the long term compared to using multiple distinct methods to detect different types of variations separately [37].

Whole-genome sequencing (WGS) is a critical genomic tool that provides comprehensive genomic information. WGS involves the initial sequencing of the complete genome of a species or individual, aiming to construct a reference genome for the species [38]. Data interpretation relies on de novo assembly, which requires overcoming technical challenges such as repetitive sequences and complex genomic regions (e.g., high GC content or polyploid genomes) [39]. However, since WGS typically targets only a few individuals (or even a single individual), its limitation lies in its inability to directly reflect genetic diversity within populations. Although it provides essential foundational data for subsequent research, it struggles to deeply analyze genetic differences among individuals within a population or systematically compare genetic differentiation between populations. In contrast, WGRS effectively addresses these shortcomings. By aligning sequencing data to a reference genome, WGRS can accurately identify various types of genetic variations, including SNPs, InDels, and SVs [40]. This makes WGRS a powerful tool for studying population genetic structures, identifying functional genes, and conducting trait association analyses, thereby offering deeper insights into genetic differences and evolutionary mechanisms within and between populations.

In conclusion, WGRS, SNP arrays and WGS play indispensable roles in genomics research, each tailored to specific research needs and objectives. WGRS excels in providing comprehensive genomic insights, offering high resolution, extensive coverage, and the ability to detect a wide range of genetic variations, making it particularly valuable for studies requiring detailed genomic analysis and low-input samples.

## 3. Applications and Research Progress in Ruminants

### 3.1. Population Genetic Structure Analysis

Population genetic structure analysis in ruminants focuses on examining genetic differences among populations and understanding the historical dynamics of populations. By analyzing the genetic structure of these economically important animals, researchers can uncover their evolutionary history, population differentiation, and adaptation to diverse environmental conditions [41]. These insights are crucial for guiding population management and breeding programs, conserving genetic resources, and improving production performance and adaptability in ruminants.

#### 3.1.1. Genetic Structure of Sheep and Goats

Recent studies on the genetic structure of sheep and goats have revealed genetic similarities, levels of differentiation, and population dynamics among various breeds.

Whole-genome sequencing of six Tibetan sheep breeds was performed using Illumina technology, and the obtained genomic data were aligned to the reference genome to identify two types of genetic variations, SNPs and InDels. Phylogenetic tree analysis based on the genetic variation data revealed genetic differentiation among these breeds, although the degree of divergence was relatively low, with extensive gene flow observed between populations. These sheep breeds were classified into three ecological types: plateau type, valley type, and Oula type [42]. Integrating genomic and transcriptomic analyses indicated that the Lanping black-boned sheep originated from the Lanping ordinary sheep and has evolved into a distinct breed [43]. A study that compared resequencing data from eight sheep breeds, involving 28 individuals, with the reference genome found that Tong sheep clustered with coarse-wool breeds such as Hu sheep, small-tailed Han sheep, Lop sheep, and Ujumqin sheep, while Hanzhong sheep showed greater genetic distance from Tong sheep. Principal component analysis (PCA) and ADMIXTURE results indicated that Tong sheep and other coarse-wool breeds belong to the Chinese Mongolian/Kazakh sheep lineage, whereas Hanzhong sheep maintained a relatively isolated breeding environment and genetic background due to its unique geographical location [44]. Further analysis of four sheep populations (European, Yunnan, Kazakh, and Mongolian) revealed that Tibetan sheep (GBF) clustered with Kazakh and Mongolian sheep in PCA but showed genetic contributions from Yunnan sheep in population structure analysis. Gene flow between Tibetan sheep and other populations was likely influenced by historical breeding practices and geographical proximity [45]. Studies have classified domestic sheep into African indigenous, Chinese indigenous, and improved breeds. Chinese indigenous breeds are further divided into Mongolian, Kazakh, and Tibetan sheep groups, exhibiting distinct geographical distributions and morphological characteristics [46].

In goats, population genetic structure analyses have shown high genetic similarity between Wuxue goats and Guizhou White goats, Chengdu Brown goats, and Jintang black goats, with significant genetic differentiation from other goat breeds [47]. Analysis of 20 Longlin goats and 66 genomes from 7 global breeds revealed that Longlin goats exhibit low linkage disequilibrium and large effective population size [48]. Genomic studies of 30 Guizhou black goats indicated shared ancestry with Shaanbei white cashmere, Yunshang black, Iran indigenous, and Moroccan goats. Guizhou black goats demonstrated significantly higher genetic diversity and lower linkage disequilibrium than other goat breeds [49]. Research on highland and lowland Leizhou goats revealed significant genetic differentiation between the two groups. Genome-wide association studies (GWASs) identified significant associations between genetic variations on NC_030818.1 and leg length traits [50]. Xiao et al. [51] successfully assembled the goat Y chromosome and analyzed resequencing data from 96 global goats. Their findings revealed two major paternal lineages (Y1 and Y2) in pre-domestication goats, a maternal bottleneck effect 2000 years ago, and global dispersal of A haplogroup goats from the Near East. Evidence from Y chromosomes suggested unbalanced gender contributions during goat trade and population expansion after the Neolithic period.

#### 3.1.2. Genetic Structure of Bovine

Recent studies have provided deeper insights into the genetic structure and population differentiation among various cattle breeds. Analysis of resequencing data from 14 Chinese cattle breeds revealed a gradient distribution of mixed Chinese taurine and indicine cattle from northern to southern China [52]. To investigate the genetic composition of Xinjiang brown cattle, genomes from 50 individuals were compared with those of eight global breeds. Results showed that Xinjiang brown cattle inherited 9.88% of their genetic material from Kazakh cattle, contributing environmental adaptation genes, and 90.12% from Swiss brown cattle, providing production-related genes [53]. Bohai black cattle exhibited mixed taurine and indicine ancestry, with close genetic relationships to Jiaxian red and Luxi cattle, and showed high genetic diversity [54]. Genome analysis of 29 Loudi cattle and 96 individuals from global cattle breeds indicated that Loudi cattle originated from East Asian, Chinese, European taurine, and Indian taurine lineages [55]. In order to explore the genetic structural relationship between the Pinan cattle population and other populations, phylogenetic trees, clustering analysis, and population structure analysis were performed. It is more similar to the northern Chinese breeds (Zhangmu cattle, Anxi cattle, Qaidam cattle, Yanbian cattle, and Xinjiang Brown cattle), and significantly different from the southern Chinese breeds (Leiqiong cattle, Wannan cattle, Wenshan cattle, Xiangxi cattle, and Dianzhong cattle) [56].

In yaks, genetic structure analysis of 198 individuals from 6 breeds (Muli, Jinchuan, Changtai, Maiwa, Zhongdian, and Tibetan) revealed that Muli yaks formed a distinct population, while Maiwa yaks exhibited complex genetic structure with gene flow from Jinchuan and Changtai yaks [57]. Molecular marker identification and population structure analysis of Subei yaks indicated they form an independent group within Chinese yak populations [58]. Resequencing data from 91 domestic yaks and 1 wild yak suggested that yak domestication likely originated in the southeastern Qinghai–Tibet Plateau before spreading westward and northeastward, with breed distribution closely linked to historical trade routes and gene flow among domestic, wild, and Tibetan yaks [59]. Single-cell RNA sequencing and resequencing of Tianzhu white yaks revealed significant genetic differentiation from other populations, while gene flow was observed among Pali and Sibu yaks in high-altitude regions [60].

These findings enhance our understanding of the genetic background of cattle breeds and provide valuable guidance for future conservation, genetic resource utilization, population improvement, and adaptive breeding programs.

### 3.2. Assessment of Genetic Diversity

In recent years, the rapid progression of industrialization and urbanization has significantly impacted natural environments, posing unprecedented challenges to the genetic diversity of many ruminant species. Some rare breeds of sheep, goats, and cattle are facing population declines and even endangerment due to a sharp reduction in genetic diversity [61]. In this context, precise and comprehensive assessments of genetic diversity in ruminants have become urgent. Understanding the distribution of gene frequencies and genetic variation levels can provide a scientific foundation for developing conservation strategies for endangered ruminants [62]. These efforts not only contribute to biodiversity conservation but also offer valuable insights for optimizing existing breeds and developing new breeds in animal husbandry.

#### 3.2.1. Genetic Diversity in Sheep and Goats

Recent studies have analyzed the genetic diversity of 11 Tibetan sheep populations, revealing low genetic diversity, moderate genetic differentiation, and a gradual increase in effective population size over time [63]. Comparative genomic analysis of Chaka sheep (CKA) with other Chinese sheep breeds (Bayinbuluke sheep, Tan sheep, and Oula sheep) was conducted, incorporating the length of runs of homozygosity (ROH) and linkage disequilibrium (LD) decay. The results revealed that the genetic diversity of CKA and TAN was significantly higher than that of OLA and BYK. Furthermore, the inbreeding coefficient (FROH) calculated based on ROH indicated lower inbreeding levels in CKA and Tan, suggesting milder inbreeding depression and relatively better genetic health status [64]. Analysis of 35 Liangshan semi-fine-wool sheep (LSS) using genomic relationship matrices (GRM) indicated low genetic relatedness among most individuals (genetic relatedness based on SNPs < 0.05). The observed heterozygosity (Ho) and expected heterozygosity (He) across all SNP loci were 0.330 and 0.334, respectively, indicating high genetic diversity in the breed [65].

Comparative genomic analyses of Xiangdong black goats and six other Chinese goat breeds showed moderate genetic diversity, low inbreeding, and large effective population sizes in Xiangdong black goats, reflecting a random mating pattern within the population [66]. Genomic analyses of 16 Hainan black goats and 71 individuals from six other geographically distinct goat breeds revealed that the genetic diversity of Hainan black goats was closely linked to their geographic distribution. PCA and phylogenetic tree analysis showed that Hainan black goats were genetically similar to Leizhou goats, consistent with their geographical proximity [67]. Further analysis of SNP density, ROH, iHS, effective population size, and nucleotide diversity in Hainan black goats identified 23,608,983 SNPs. ROH analysis suggested a moderate level of inbreeding in this breed [68]. Genomic resequencing of 123 cashmere goats assessed inbreeding levels, revealing that Inner Mongolian cashmere goats had the lowest inbreeding coefficient (0.0263). A total of 57,224 ROHs and 74 ROH islands were identified [69]. These studies highlight the complex genetic relationships and population histories among different goat breeds, emphasizing the importance of conserving genetic diversity and providing valuable genetic information for future improvement and management of sheep and goat populations.

#### 3.2.2. Genetic Diversity in Bovine

Genetic diversity studies of Yunling cattle revealed low levels of nucleotide diversity [70]. Analysis of Qinchaun cattle showed high genetic diversity, identifying over 20 million SNPs and an IBS distance of 0.243. Some individuals exhibited close genetic relationships. Additionally, 8258 ROHs were detected, and the inbreeding coefficient was calculated as 0.039, indicating moderate inbreeding, especially in bulls [71]. Genome analysis of 20 Kashmir cattle and other breeds showed low inbreeding and high nucleotide diversity, reflecting strong genetic diversity in Kashmir cattle [72]. Genetic diversity assessments of Nanyang cattle revealed closer genetic relationships with central and northern Chinese cattle breeds than with European breeds, higher genetic diversity, and low inbreeding levels. Selection signals related to reproduction and immunity were also identified [73]. A comparative analysis of 23 Xiangxi cattle and 78 genomes from 6 globally representative breeds revealed weak artificial selection pressures and high genomic diversity in Xiangxi cattle, as shown by nucleotide diversity, inbreeding coefficients, linkage disequilibrium decay, and ROH analyses [74]. Comparison of 23 Zaobei cattle with 46 Simmental cattle demonstrated clear genetic separation between the two breeds, confirming the genetic uniqueness of Zaobei cattle [75].

These whole-genome resequencing studies comprehensively evaluated the genetic diversity and inbreeding levels of various cattle breeds, uncovering their unique genetic characteristics and adaptive traits. These findings provide crucial insights for genetic resource conservation and breed improvement in cattle populations.

### 3.3. Identification of Functional Genes Associated with Economic Traits

In modern animal husbandry, ruminants are among the most important livestock species, and improving their economic traits—such as wool yield, meat quality, reproductive performance, and growth rate—is crucial for enhancing productivity and economic returns [76]. The genetic basis of these traits is directly linked to breed improvement and the sustainable development of the industry. Advances in molecular biology, particularly genomics and bioinformatics, have enabled the identification and understanding of key functional genes underlying these important economic traits [77].

#### 3.3.1. Key Functional Genes in Sheep and Goats

Recent whole-genome resequencing (WGRS) studies on sheep and goats have revealed genomic regions and candidate genes associated with key economic traits, including reproduction, growth, and immune response. By aligning the Hu sheep genome to the reference genome, a total of 39,467,233 SNPs and 8,677,193 insertions/deletions (InDels) were identified. Selection signature analysis revealed genetic selection signals associated with litter size, pinpointing the *CC2D1B* gene and other reproduction-related genes. Additionally, specific missense mutations and linkage disequilibrium patterns were detected in Hu sheep [78]. In Hu sheep, missense mutations in *GPR35* and *NAV1* were significantly associated with immunity and growth traits [79]. Growth-related candidate genes such as *HDAC1*, *ACVR1*, and *GNAI2* in Hu sheep, and *RBBP8*, *PLAT*, and *CRB1* in Gangba sheep, were identified. Both populations showed strong selection for the *CYP2E1* gene, associated with growth [80]. In fine-wool sheep, selective scans in horned and polled groups revealed strong selection on the *RXFP2* gene [81]. Wool and cashmere fibers are highly valued for their unique properties and play a significant role in various industries. Studies have shown that wool-related traits exhibit substantial heritability, indicating strong genetic control over these traits and the potential for improvement through selective breeding. This finding provides a critical theoretical foundation for the genetic enhancement and breeding of wool traits [82]. Genome-wide association studies (GWASs) on wool traits in Alpine Merino, Chinese Merino, Qinghai fine-wool sheep, and Aohan fine-wool sheep identified SNPs associated with wool length, fiber diameter, greasy fleece weight, and clean fleece yield, along with estimates of heritability for these traits [83]. Comparative resequencing of wild and domesticated sheep revealed nonsynonymous mutations in candidate genes, with *PDGFD* emerging as a key gene influencing tail fat deposition in sheep [84]. Researchers conducted resequencing and genome-wide association analysis on Ujimqin sheep with multiple vertebrae and determined that the *ABCD4* gene is associated with the number of vertebrae in sheep. The transcriptome expression profile of developing mouse embryos shows that *ABCD4* is highly expressed during the critical period of vertebral body formation (4.5–7.5 days). The *ABCD4* gene has two missense mutations, and the mutation at position 22 (Chr7: 89393414, C>T) converts arginine to glutamine, which hinders the function of the protein in vertebral development [85].

In dairy and non-dairy goat breeds, WGRS analysis of 89 individuals identified candidate genes related to milk production (*GHR*, *DGAT2*), reproduction (*PTGS2*, *ESR2*), and immunity (*POU2F2*, *LRRC66*) [86]. Genomic analysis of eight local and imported dairy goat breeds revealed key candidate genes for milk production (*STK3*, *PRELID3B*), reproduction (*ATP5E*), growth (*CTSZ*, *GHR*), and immune function (*CTSZ*, *NELFCD*) [87]. GWAS in Dazu black goats, known for high reproductive performance, identified reproduction-related genes such as *ATP1A1* and *FGFR2* (udder traits), and *GASK1B* and *ENSCHIG00000026285* (litter size) [88]. Genomic studies of Lanzhou fat-tailed sheep identified genes associated with androgen synthesis and follicle-stimulating hormone response (*SRD5A2*, *SRD5A3*, and *PAWR*) that may contribute to intersex traits. Copy number variations (CNVs) in chromosomes 1, 4, 9, and 16 were also implicated in the expression of genes influencing these traits [89]. Analysis of six multi- and single-kid female goats identified reproductive candidate genes such as *AURKA*, *ENDOG*, *CDC25C*, and *NANOS3*, which are mainly involved in protein coiled-coil structure and ovarian development [90].

These findings provide valuable molecular markers and candidate genes for genetic improvement and breeding programs in sheep and goats, enhancing our understanding of the mechanisms underlying key economic traits such as reproduction, milk production, and immunity.

#### 3.3.2. Key Functional Genes in Bovine

Elucidating the functions and regulatory mechanisms of genes associated with economic traits in cattle is of significant importance for gaining a deeper understanding of the genetic basis of trait formation and advancing molecular breeding practices in cattle. In Dabieshan cattle, selective scans identified candidate genes associated with reproduction (*GPX5*, *GPX6*), feed efficiency (*SLC2A5*), immune response (*IGLL1*), heat tolerance (*DnaJC1*), fat deposition (*MLLT10*), and coat color (*ASIP*) [91]. For Xianan cattle, a hybrid of Charolais and Nanyang breeds with robust body size and high meat yield, analysis of 30 individuals and 178 published genomes revealed candidate genes related to skeletal development (*NR6A1*), meat quality (*MCCC1*), growth (*WSCD1*), and immunity (*IL11RA*, *GOLM1*) [92]. Annotation of 20 Chaling cattle genomes identified strong selection signals in genes related to immunity, heat tolerance, reproduction, growth, and meat quality [93]. Genomic analysis of Yunling cattle identified 3728 CNV regions, 111 of which overlapped with 76 quantitative trait loci (QTLs) for traits such as subcutaneous fat thickness, longissimus dorsi muscle area, and marbling score [94].

Analysis of whole-genome resequencing data from Hanwoo cattle identified a series of candidate genes that may play critical regulatory roles in milk production, meat quality traits (including tenderness, juiciness, and marbling), and the genetic mechanisms underlying the yellow coat color phenotype [95]. CNV analysis across major water buffalo breeds worldwide mapped CNV regions associated with growth, body weight, and horn development [96]. In yaks, a vital resource on the Qinghai–Tibet Plateau, genomic studies identified 203 selection regions and 27 genes linked to traits such as short stature, milk quality, meat quality, fertility, adaptability, growth, and immunity [97]. Comparative analysis of long-haired and normal-haired Tianzhu white yaks identified 80 genes associated with long-hair traits, including those related to lipid metabolism and cell migration, such as *COL22A1*, *GK5*, and *SLIT3* [98]. The Jinchuan yak, known for its additional thoracic vertebrae, revealed 330 markers associated with this trait. Validation in 51 yaks confirmed 7 markers with high predictive accuracy. Candidate genes *PPP2R2B* and *TBLR1* were identified for the unique vertebral number [99]. These findings provide a robust scientific basis for future cattle breed improvement and genetic breeding strategies. A summary of candidate genes associated with economic traits in ruminants are presented in Table 1.

### 3.4. Studies on Adaptive Traits

Understanding the mechanisms underlying the adaptation of cattle and sheep to extreme climates, high altitudes, and various pathogens is crucial for guiding breeding programs, optimizing husbandry practices, and safeguarding the health and welfare of these populations. Research on adaptive traits not only enhances the understanding of the biology of livestock but also provides a scientific basis for addressing challenges posed by climate and environmental changes, supporting global agriculture and environmental management [100].

#### 3.4.1. Adaptive Traits in Sheep and Goats

Recent studies have investigated the genetic adaptation of Chinese indigenous sheep to different climatic conditions, identifying candidate genes associated with adaptation to extreme environments. By aligning to the sheep reference genome and conducting genomic analyses of Kazakh sheep (KAZ), Mongolian sheep (MON), Tibetan sheep (TIB), and Yunnan sheep (YUN), candidate genes associated with adaptation to arid regions (*TBXT*, *TG*, *HOXA1*), high-altitude environments (*DYSF*, *PDGFD*, *NF1*), and warm climates (*TSHR*, *ABCD4*, *TEX11*) were identified [101]. Analysis of copy number variations (CNVs) in 24 Tibetan sheep identified variations in the RUNX1 gene, which is associated with hypoxic adaptation, enabling Tibetan sheep to thrive in the low-oxygen environment of the Qinghai–Tibet Plateau [102]. *RUNX1* plays essential roles in biological processes such as bone development [103], immune cell development [104], and hypoxia-related gene expression regulation [105], highlighting its significance in high-altitude adaptation. Further research identified 338 breed-specific selective regions in Chinese indigenous sheep, including *CYP17*, associated with hypoxic adaptation in Tibetan sheep, and *DNAJB5*, related to heat tolerance in Duolang sheep [106]. In Hainan black goats, 33 candidate genes were linked to heat tolerance (*GNG2*, *MAPK8*) and stress resistance (*TLR2*, *IFI44*, *ENPP1*) [68].

#### 3.4.2. Adaptive Traits in Bovine

Extensive studies have explored the genetic basis of adaptation in cattle to high-altitude and extreme climatic conditions, identifying genetic markers and candidate genes involved in these processes. Genomic analysis of nine Yanbian cattle identified cold tolerance genes such as *CORT*, *FGF5*, and *CD36* [107]. Comparison of CNVs in Yakut cattle and Holstein cattle with those of Korean and Holstein breeds revealed CNVRs associated with cold adaptation [108]. Genomic analysis of Gansu Meiren yaks identified candidate genes such as *BMP2* and *ACVR2B*, related to high-altitude and hypoxic adaptation [109].

Analysis of genomes from 32 breeds, including 258 cattle, revealed introgression signals between yaks and Tibetan cattle involving genes linked to hypoxia response (*EGLN1*), cold adaptation (*LRP11*), DNA damage repair (*LATS1*), and UV radiation resistance (*GNPAT*). SNPs in the *EGLN1* promoter region reduced its expression, enhancing hypoxia tolerance in Tibetan cattle [110]. SNP functional annotation in Siberian cattle identified chromosomal regions and candidate SNPs associated with thermoregulation and genes linked to heat adaptation, such as *GRIA4* and *COX17* [111]. Genomic analysis of Lincang humped cattle, which contain indicine and taurine ancestry, identified a missense mutation in *HELB* related to heat tolerance, providing insights into indicine cattle’s adaptation to hot environments [112].

CNV and CNVR analyses in Chinese, Indian, and European commercial cattle breeds identified genes associated with environmental adaptation (*CNGB3*, *FAM161A*, *ERC2*) and oxidative stress and anti-inflammatory responses (*COMMD1*, *OXR1*) [113]. Comparative genomic analysis of Xinjiang Mongolian cattle and 134 genomes from 9 representative breeds worldwide revealed candidate genes related to desert adaptation, including genes involved in water reabsorption, osmotic regulation, metabolism, and energy balance [114]. Genotyping and resequencing of Chinese indigenous cattle genomes revealed candidate genes related to thermogenesis and energy metabolism, such as *UQCR11*, *EGR1*, and *STING1* [115]. Studies of Iran indigenous river buffalo populations identified a region on chromosome 2 overlapping the *OCA2-HERC2* gene, indicating a genetic basis for UV adaptation and pigmentation mechanisms in water buffalo [116]. Genomic analysis of 25 domesticated buffalo breeds revealed selection signals in swamp buffalo related to neural system genes and in river buffalo associated with heat stress and immune genes [117]. Comparison of salt-tolerant Chilika buffalo and non-salt-tolerant Murrah buffalo identified pathways related to salt tolerance, including MAPK signaling and renin secretion [118]. Genomic analysis of Xiangxi white buffalo (XWB) identified candidate genes associated with nervous system traits (*GRIK2*), reproduction (*KCNIP4*), growth and development (*IFNAR1*), morphology (*LINGO2*), and immunity (*IRAK3*) [119].

These findings deepen the understanding of the genetic basis of adaptation in cattle breeds and provide critical scientific tools for future genetic improvement and breeding programs focused on environmental adaptability. A summary of candidate genes associated with adaptive traits in ruminants can be found in Table 2.

## 4. Conservation of Genetic Resources

Ruminants, a vital component of global agriculture, play a crucial role in ensuring food security, maintaining ecological balance, and supporting rural livelihoods. They provide us with essential products such as meat, milk, and hides, while also contributing to soil fertility and nutrient cycling [120]. However, the genetic resources of these invaluable animals are under increasing threat. The widespread adoption of a few high-yielding breeds has led to the erosion of genetic diversity, leaving ruminant populations vulnerable to diseases, environmental changes, and market fluctuations. The loss of indigenous breeds, each uniquely adapted to specific environments and production systems, further exacerbates this problem [121]. The conservation of ruminant genetic resources is thus an urgent task. In the face of such a challenging situation, the whole-genome resequencing (WGRS) technology has emerged as a ray of hope.

WGRS technology has played a pivotal role not only in advancing the study of genetic diversity and gene function but also in the development, conservation, and utilization of genetic resources [122]. This technology enables detailed analysis of species’ genetic diversity and provides a powerful tool for monitoring and protecting biodiversity. By examining genetic structures and population histories, WGRS facilitates the elucidation of species’ evolutionary processes, offering scientific insights for the conservation and management of endangered species.

### 4.1. Conservation of Genetic Resources in Sheep and Goats

The Inner Mongolian Cashmere goat (IMCG), a premium cashmere breed in China, has been the focus of conservation efforts. To effectively preserve and utilize the purebred genetic resources of IMCG, researchers performed WGRS on 225 randomly selected individuals. The results showed low levels of runs of homozygosity (ROH), a low mean inbreeding coefficient, and high genetic diversity within the IMCG population [123]. In Tibetan sheep, WGRS analyses of 11 populations indicated low genetic diversity, minimal genetic differentiation, and clear population structures. Effective population sizes increased over time, and evolutionary relationships among populations were closely associated with geographical and genetic distances, reflecting the profound influence of historical activities on Tibetan sheep populations [124]. Genomic data analysis of Baiyu black goats and Chuanzhong black goats was conducted to assess genetic diversity, population structure, and selection signatures. The results revealed that Baiyu black goats exhibit greater genetic diversity and significant genetic differentiation compared to Chuanzhong black goats. Phylogenetic analysis indicated that Baiyu black goats are more closely related to Tibetan cashmere goats, while Chuanzhong black goats show closer genetic affinity to Chengdu gray goats. These findings provide a theoretical foundation for the further conservation of genetic resources in Baiyu black goats and Chuanzhong black goats [125].

### 4.2. Conservation of Genetic Resources in Bovine

Kazakh cattle, known for their adaptability and versatility in Xinjiang, were analyzed to study their evolutionary history, breeding characteristics, and germplasm resource conservation. Researchers compared genomes from 26 Kazakh cattle with 103 genomes from 7 other global cattle breeds. The findings revealed large effective population sizes, low linkage disequilibrium, and a population structure predominantly derived from European taurine ancestry [126]. Kongshan cattle, recognized for their resilience in harsh environments, have experienced population declines due to competition with exotic breeds. Using WGRS, researchers identified substantial genetic variation in this breed, including mutations in NEIL2 and PNKP, which are associated with stress resistance. These findings are instrumental in conserving the genetic diversity of Kongshan cattle [127]. To conserve and utilize the endangered Dengchuan cattle, a unique breed in Yunnan Province, researchers sequenced 10 individuals. The results revealed a high level of hybridization, with positive selection signals associated with milk production, disease resistance, growth, and heat tolerance. These findings provide a theoretical basis for understanding the genetic mechanisms underlying Dengchuan cattle’s desirable traits, such as high milk yield, adaptability, roughage tolerance, immune performance, and small body size [128].

In conclusion, the application of WGRS has not only deepened the understanding of genetic diversity and population structures in cattle and sheep but has also provided a robust tool for the effective conservation and sustainable utilization of genetic resources. This technology holds significant scientific value for maintaining biodiversity and promoting sustainable agricultural development.

## 5. Deciphering Complex Traits in Ruminants Using Whole-Genome Resequencing: Challenges and Multi-Omics Breakthroughs

Whole-genome resequencing (WGRS) has played a significant role in the analysis of genetic diversity, genomic selection [129], disease research [130], and genetic resource conservation in ruminants, providing a powerful tool for ruminant genetics and breeding research. However, despite its notable advantages in many aspects, WGRS still faces certain limitations in deciphering complex traits.

Complex traits are influenced by multiple genes and environmental factors, and their inheritance patterns do not conform to classical Mendelian genetics [131]. Unlike simple traits controlled by single genes, complex traits typically exhibit quantitative characteristics, such as milk yield [132], growth rate [133], and wool traits [134] in ruminants. The genetic basis of these traits involves multiple genes, each with small effects, and may include gene–gene interactions (epistasis) and gene–environment interactions. Furthermore, the phenotypic variation of complex traits is often continuous, making it difficult to interpret through simple genetic models [135]. Although WGRS can comprehensively detect genomic variations, it still encounters challenges in unraveling the polygenic regulatory networks, gene interactions, and environmental influences underlying complex traits.

To fully elucidate the genetic mechanisms of complex traits, WGRS needs to be integrated with multiple technologies to construct a multi-level research framework. Transcriptome sequencing (RNA-seq) serves as a crucial complementary technology, linking genomic variants identified by WGRS to functional gene expression changes and revealing the impact of variants on gene regulation [136]. Epigenomic technologies, such as DNA methylation sequencing and chromatin accessibility analysis, can further dissect the role of epigenetic modifications in complex traits, elucidating how environmental factors influence gene expression through epigenetic regulation [137]. Additionally, metabolomics and proteomics provide insights into downstream metabolic products and protein levels, helping to construct a comprehensive regulatory network from genomic variation to phenotypic manifestation [138]. In terms of data analysis, integrating systems biology approaches and machine learning algorithms with multi-omics data enables a more comprehensive understanding of the contributions of gene–gene and gene–environment interactions to complex traits [139].

By combining WGRS with transcriptomics, epigenomics, metabolomics, proteomics, and other multi-omics technologies, and complementing these with systems biology analysis and functional validation experiments, the genetic basis of complex traits can be fully deciphered. This integrated approach provides a scientific foundation for genetic improvement and precision breeding in ruminants.

## 6. Future Perspectives

With the continuous advancement of high-throughput sequencing technologies and declining sequencing costs, WGRS has gained widespread application. Given the massive datasets generated by resequencing, future research will increasingly depend on the development of advanced bioinformatics tools and algorithms to efficiently and accurately interpret these data, enhancing the efficiency and precision of data analysis [140]. Improvements in computing power, memory, storage technologies, and data processing capabilities have driven the adoption of machine learning algorithms in biomedical research [141], and these algorithms also show significant potential in WGRS studies of ruminants. For instance, clustering algorithms can efficiently classify and organize resequencing data, rapidly identifying characteristic gene sequences in different ruminant populations, thus providing more efficient methods for analyzing population genetic structures [142]. Deep learning algorithms, by learning from large datasets of known functional gene–trait associations, can establish precise predictive models. These models have the potential to accurately predict the functions and roles of unknown genes in economic and adaptive traits, significantly reducing research timelines [143].

While WGRS focuses on analyzing genetic differences at the genomic level, multi-omics (integrating genomics, epigenomics, transcriptomics, proteomics, and metabolomics) provides a systematic approach to understanding complex biological phenomena and mechanisms by combining multiple layers of data [144]. For example, common variable immunodeficiency (CVID), caused by multigenic variations and characterized by diverse clinical manifestations, can be better understood through multi-omics analysis, revealing the underlying molecular complexity of the disease [145]. The integration of WGRS and multi-omics analysis enables efficient identification of key genetic information and systematic elucidation of the molecular regulatory networks and genetic mechanisms underlying complex traits, thereby advancing livestock breeding programs. With continuous theoretical advancements and practical innovations, significant breakthroughs in this field are anticipated in the near future.

## 7. Conclusions

This review provides a concise overview of the principles and advantages of whole-genome resequencing (WGRS) technology and systematically explores its applications and recent advancements in ruminant research. In the section on “Applications and Research Progress in Ruminants,” it highlights the use of WGRS for analyzing population genetic structures, elucidating the genetic relationships and evolutionary history among different ruminant populations. Through the assessment of genetic diversity, it further clarifies the genomic diversity characteristics and genetic backgrounds of various ruminant breeds. Additionally, the review delves into the application of WGRS in identifying functional genes associated with economic traits (such as meat quality and milk yield) and adaptive traits, emphasizing its critical role in enhancing livestock production efficiency and breeding precision. However, despite the significant advantages of WGRS in detecting genomic variations, it still faces challenges and limitations in deciphering the polygenic regulatory networks and gene–environment interactions underlying complex traits.

## Figures and Tables

**Table 1 animals-15-00831-t001:** Correspondence table of economic trait candidate genes.

Traits	Bovidae		Candidate Gene	Reference
	Bovina	Caprinae		
Reproduction	Dabieshan Cattle, Huanhu yaks, Maiwa yaks, Yushu yaks	Hu Sheep, Saanen, Nubian, Alpine, Toggenburg, Guanzhong dairy goat	*BMPR1B*, *BMP2*, *PGFS*, *CYP19*, *CAMK4*, *GGT5*, *GNAQ*, *TSHR*, *TSHB*, *PTGS2*, *ESR2*, *ATP5E*, *AURKA*, *ENDOG*, *SOX2*, *RORA*, *GJA10*, *RXFP2*, *CDC25C*, *NANOS3*, *GPX5*, *GPX6*, *BTBD11*, *ARFIP1*	[79,86,87,90,91,97]
Vision		Hu Sheep	*ALDH1A2*, *SAG*, *PDE6B*	[79]
Immunity	Dabieshan Cattle, Xia’nan cattle, Huanhu yaks, Maiwa yaks, Yushu yaks	Hu Sheep, Saanen, Nubian, Alpine, Toggenburg, Guanzhong dairy goat, Xinong Saanen Dairy Goat, Laoshan Dairy Goat	*GPR35*, *SH2B2*, *PIK3R3*, *HRAS*, *JAK1*, *POU2F2*, *LRRC66*, *CTSZ*, *NELFCD*, *IGLL1*, *BOLA-DQA2*, *BOLA-DQB*, *IL11RA*, *CNTFR*, *CCL27*, *SLAMF1*, *SLAMF7*, *NAA35*, *GOLM1*, *INPP5D*, *ADCYAP1R1*	[79,86,87,91,92,97]
Growth	Xia’nan cattle, Huanhu yaks, Maiwa yaks, Yushu yaks	Gangba sheep, Hu Sheep, Saanen, Nubian, Alpine, Toggenburg, Guanzhong dairy goat, Xinong Saanen Dairy Goat, Laoshan Dairy Goat	*HDAC1*, *YH7B*, *LCK*, *ACVR1*, *GNAI2*, *RBBP8*, *ACSL3*, *FBXW11*, *PLAT*, *CRB1*, *CTSZ*, *GHR*, *NR6A1*, *WSCD1*, *TMEM68*, *MFN1*, *NCKAP5*, *GRHL2*, *GRID2*, *SMARCAL1*, *EPHB2*	[80,87,92,97]
Yearling Staple Length		Alpine Merino sheep, Chinese Merino sheep, Qinghai fine-wool sheep, Aohan fine-wool sheep	*FAM46A*, *LOC105607652*, *SGCD*, *PLCE1*, *RAB3C*, *AIM1*, *FAT3*, *C6H4orf22*, *PLA2R1*, *ANKRD42*, *KANSL2*, *ZNF804A*, *DPYD*, *LOC101114228*, *LOC101122569*, *LOC105607993*, *PLA2R1*, *LOC105611292*, *PRTFDC1*, *LOC101122517*, *C21H11orf85*, *BATF2*	[83]
Yearling Mean Fiber Diameter		Alpine Merino sheep, Chinese Merino sheep, Qinghai fine-wool sheep, Aohan fine-wool sheep	*LOC106990409*, *ARID1A*, *MGMT*, *U2AF1*, *CRYAA*, *SLIT3*, *ARHGAP15*, *N4BP2*, *TRNAG-UCC*, *LOC105611432*, *LOC105603404*, *SLC4A11*, *LOC105603112*, *LOC105606895*, *LOC101115632*	[83]
Milk Production	Ankole, Kenana, Huanhu yaks, Maiwa yaks, Yushu yaks	Saanen, Nubian, Alpine, Toggenburg, Guanzhong dairy goat, Laoshan Dairy Goat	*GHR*, *DGAT2*, *ELF5*, *GLYCAM1*, *ACSBG2*, *ACSS2*, *STK3*, *PRELID3B*, *HECW1*, *HECW2*, *OSBPL2*	[86,87,97]
Udder Traits		Dazu Black Goat	*ATP1A1*, *LRRC4C*, *SPCS2*, *XRRA1*, *CELF4*, *NTM*, *TMEM45B*, *ATE1*, *FGFR2*	[88]
Litter Size		Dazu Black Goat, Shaanbei White Cashmere Goat	*ENSCHIG00000017110*, *SLC9A8*, *GLRB*, *GRIA2*, *GASK1B*, *ENSCHIG00000026285*, *CDC25C*, *ENDOG*, *NANOS3*	[88,90]
Meat	Xia’nan cattle, Huanhu yaks, Maiwa yaks, Yushu yaks		*MCCC1*, *BZW1*, *AOX1*, *LOC100138449*, *SPATA5*, *GRHL2*	[92,97]
Hair Growth And Hair-Follicle Development	Tianzhu white yaks		*ASTN2*, *ATM*, *COL22A1*, *GK5*, *SLIT3*, *PM20D1*, *SGCZ*	[98]

**Table 2 animals-15-00831-t002:** Correspondence table of adaptive trait candidate genes.

Adaptive Traits	Bovidae		Candidate Gene	Reference
	Bovina	Caprinae		
Drought-prone		Kazakh sheep, Mongolian sheep, Tibetan sheep, Yunnan sheep,	*TBXT*, *TG*, *HOXA1*	[101]
High-altitude		Kazakh sheep, Mongolian sheep, Tibetan sheep, Yunnan sheep	*DYSF*, *EPAS1*, *JAZF1*, *PDGFD*, *NF1*	[101]
Hypoxia	Tibetan cattle	Tibetan sheep	*RUNX1*, *CYP17*, *MBP2*, *ACVR2B*, *EGLN1*	[102,106,110]
Heat tolerance	Siberian cattle, Lincang humped cattle	Duolang sheep, Hainan black goat, Kazakh sheep, Mongolian sheep, Tibetan sheep, Yunnan sheep,	*DNAJB5*, *GNG2*, *MAPK8*, *CAPN2*, *SLC1A1*, *LEPR*, *GRIA4*, *COX17*, *MAATS1*, *UPK1B*, *IFNGR1*, *DDX23*, *PPT1*, *THBS1*, *CCL5*, *ATF1*, *PLA1A*, *PRKAG1*, *NR1I2*, *MED16*, *DNAJC8*, *HSPA4*, *FILIP1L*, *HELB*, *BCL2L1*, *TPX2*, *COMMD1*, *TSHR*, *ABCD4*, *TEX11*	[68,101,106,111]
Stress resistance		Hainan black goat	*TLR2*, *IFI44*, *ENPP1*, *STK3*, *NFATC1*	[68]
Cold tolerance	Yanbian cattle, Tibetan cattle		*CORT*, *FGF5*, *CD36*, *LRP11*, *UQCR11*, *STING1*, *EGR1*	[107,110,115]
Water reabsorption and osmoregulation (desert adaptation)	Xinjiang Mongolian cattle		*PDE11A*, *DIS3L2*, *SLIT2*, *KCNIP4*, *OSGEP*, *PKHD1*, *LOC617141*, *LOC112442378*, *LOC527385*, *FECH*, *LOC530929*, *LOC511936*, *CLIC4*, *FGF10*	[114]

## Data Availability

No new data were created or analyzed in this study. Data sharing is not applicable to this article.

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
