# Peer review of "Leveraging Whole-Genome Resequencing to Uncover Genetic Diversity and Promote Conservation Strategies for Ruminants in Asia"

_animals, 2025, doi:10.3390/ani15060831_

Round 1
Reviewer 1 Report
Comments and Suggestions for Authors
Overall, the manuscript is well written and gives an overview or review of the use of WGRS in several ruminant species.
However, one major deficiency of this review is that the majority of referenced studies are on Asian breeds – noted that the are references to occasional non Asian breeds. There are many other studies performed on African, European and South American breeds which could be incorporated. I suggest that, if left as is, then the title should be modified to convey the overtly Asian bias. Table 1 for instance only shows data from Asian breeds. Table 1 should also be sub-sectioned for breed and species and referenced according to text referencing #.
Figure 1 – reference is required for this data however, I am not sure this Figure is actually required or adds anything to the review.
Questions which should be considered:
(1) How does WGRS compare to other high density snp arrays. Does WGRS give additional information? Is it cost effective?
(2) Do different methods of WGRS have advantages – such as long read NGS (eg Nanopore) versus small read NGS? Are they additive or is one better than another?
Author Response
Comments 1: However, one major deficiency of this review is that the majority of referenced studies are on Asian breeds – noted that the are references to occasional non Asian breeds. There are many other studies performed on African, European and South American breeds which could be incorporated. I suggest that, if left as is, then the title should be modified to convey the overtly Asian bias.
Response 1: “Leveraging Whole-Genome Resequencing to Uncover Genetic Diversity and Promote Conservation Strategies for Ruminants in Asia” Thank you for pointing this out.We agree with this comment. Therefore, we have revised the title of the manuscript to "Leveraging Whole - Genome Resequencing to Uncover Genetic Diversity and Promote Conservation Strategies for Ruminants in Asia". This new title clearly indicates the regional focus of our study, aligning with the content's Asian - centric nature. This new modification can be found in the title on the first page.
Comments 2: Table 1 for instance only shows data from Asian breeds. Table 1 should also be sub-sectioned for breed and species and referenced according to text referencing.
Response 2: Agree.I have added a new column for the breed. Additionally, I have revised the format of the references to match the type used in the main text. This addresses the concerns regarding Table 1, ensuring that it now includes the necessary information on breeds as suggested, and the references are in the appropriate format as per the requirements in the text. The new revisions can be found in the first table on page 9 of the article.
Comments 3: Figure 1 – reference is required for this data however, I am not sure this Figure is actually required or adds anything to the review.
Response 3: Agree. Since we were unable to find reasonable references to support Figure 1, and this figure has little relevance to the content of the article, we have decided to delete the figure.
Comments 4: How does WGRS compare to other high density snp arrays. Does WGRS give additional information? Is it cost effective?
Response 4: WGRS provides more comprehensive genomic information compared to high-density SNP arrays, particularly for rare variants and structural changes. However, SNP arrays remain more cost-effective for large-scale studies focused on known variants. The choice depends on the study's goals, budget, and the need for detailed genomic data.Thank you very much for your question. We have answered this question in the section of 2.2 Technical Advantages of the article. Please specifically refer to the ninth line on page four.
Comments 5: Do different methods of WGRS have advantages – such as long read NGS (eg Nanopore) versus small read NGS? Are they additive or is one better than another?
Response 5: Both long-read and short-read NGS have unique advantages and limitations. Long-read sequencing excels in resolving complex genomic regions and phasing, while short-read sequencing offers high accuracy and cost-effectiveness. For comprehensive WGRS, a hybrid approach leveraging both technologies can provide the most complete and accurate genomic information. The choice depends on the specific research goals, budget, and the complexity of the genomic regions being studied. Thank you very much for your question. We have answered this question in the section of 2.1 Definitions of the article. Please specifically refer to the third line on page three.
Reviewer 2 Report
Comments and Suggestions for Authors
The authors in this article have summarised the recent advances of WGRS in ruminants. The article is well prepared by adding appropriate references and sufficient information. However, the manuscript needs a major revision. Specific comments are added below.
Abstract
Please add what WGRS is, here.
The article explains the additional advantages of resequencing in ruminants. The authors need to provide an overview of the WGS in ruminants and provide a thorough explanation of the gaps in our understanding of WGS and, specifically, how resequencing fills those gaps in our understanding with specific examples.
Introduction
The introduction needs to be on the topic. The evolution of WGS in ruminants, the WGRS, and the current status of WGS in ruminants need to be introduced. The importance of WGRS also has to be introduced.
Why is only the meat production stat included in the introduction? Why not dairy and wool production? Please add these also, as the article covers ruminant production in general.
In the introduction section, pigs and poultry are also explained. The title says the review is about WGRS in ruminants. Please clarify.
What is the relevance of Figure 1 in the introduction section? From where the data were obtained for the figure. The data source needs to be specified in Figure 1.
In general, the introduction needs to be rewritten completely, considering the above.
Body of the article
2.2. Technical advantages
Be specific. What techniques were used in WGS? What were the disadvantages? How will the new techniques explained here used for resequencing resolve those shortcomings in WGS?
The body of the article is divided into subsections based on population structure, genetic diversity, functional genes studies, adaptation traits and conservation, and again subdivided based on species (Sheep and goat, and bovines). The body of the article has a good flow of information. Why are species-wise studies not included in the conservation section?
At the beginning of each sub-section, please add a description of the subsection. Please add references also. For example, in section 3.1, the first paragraph has no references. This introduction paragraph on conservation is missing in section 4, “Conservation of Genetic Resources”.
All figures need to be self-explanatory. For example, please give a brief description of the main points you want to highlight in Figure 2. Did you develop Figure 2 by yourself? If you have used the data or resources from other articles, please add those references in the Figure legend.
In each section, please explain what the main highlights generated from WGS are in these domains. For example, specify at least one example for each section. In section 3.1 Population Genetic Structure Analysis, please discuss the key points we got from WGS and how WGRS added to our current knowledge. Then, explain how resequencing helped in improving our understanding of the whole domain.
5. Conclusion and future perspective
I would add future perspective as the fifth section and add conclusion as the sixth section. Please limit the length of the conclusion section. Please summarise the key message of the article (take-home messages) in the conclusion section, avoiding references and further explanations.
References
OK.
Author Response
Comments 1: Please add what WGRS is, here.The article explains the additional advantages of resequencing in ruminants. The authors need to provide an overview of the WGS in ruminants and provide a thorough explanation of the gaps in our understanding of WGS and, specifically, how resequencing fills those gaps in our understanding with specific examples.
Response 1: We agree with this comment. We have added content about WGRS and provided an overview of the application of whole-genome sequencing (WGS) in ruminants. We have also illustrated the current deficiencies of whole-genome sequencing and explained how WGRS remedies these deficiencies. This part of the content can be found on the first page of the article.
Comments 2: The evolution of WGS in ruminants, the WGRS, and the current status of WGS in ruminants need to be introduced. The importance of WGRS also has to be introduced.
Response 2: We agree with this comment. We illustrated the evolution of whole-genome sequencing technology over time by citing examples of ruminant reference genomes. We also introduced the significance of whole-genome resequencing and whole-genome sequencing technologies. This modification can be found on the ninth line of the second page of the article.
Comments 3: Why is only the meat production stat included in the introduction? Why not dairy and wool production? Please add these also, as the article covers ruminant production in general.
Response 3: We have added data on dairy production and wool production in the introduction section. This modification can be found on the fourth line of the second page of the article.
Comments 4: In the introduction section, pigs and poultry are also explained. The title says the review is about WGRS in ruminants. Please clarify.
Response 4: We agree with this comment. Since the data on pigs and poultry do not match the title of the article, we have deleted the data related to pigs and poultry in the introduction section.
Comments 5: What is the relevance of Figure 1 in the introduction section? From where the data were obtained for the figure. The data source needs to be specified in Figure 1.
Response 5: Thank you for pointing this out. We agree with this comment. Since we were unable to find reasonable references to support Figure 1, and this figure has little relevance to the content of the article, we have decided to delete the figure.
Comments 6: What techniques were used in WGS? What were the disadvantages? How will the new techniques explained here used for resequencing resolve those shortcomings in WGS?
Response 6: Thank you for pointing this out. Therefore, we consulted the references and added content about whole-genome sequencing technology, explained its deficiencies, and also added content on how whole-genome resequencing technology remedies these deficiencies. This modification can be found on line 26 of page 4 of the article.
Comments 7: The body of the article is divided into subsections based on population structure, genetic diversity, functional genes studies, adaptation traits and conservation, and again subdivided based on species (Sheep and goat, and bovines). The body of the article has a good flow of information. Why are species-wise studies not included in the conservation section?
Response 7: Thank you for pointing this out. We also agree with this opinion. Therefore, we have reorganized the content in the conservation section according to different species. This modification can be found on the seventh line of page 14 of the article.
Comments 8: At the beginning of each sub-section, please add a description of the subsection. Please add references also. For example, in section 3.1, the first paragraph has no references.
Response 8: Thank you for pointing this out. We also agree with this opinion. Therefore, we have supplemented the description of each subsection and added references. This modification can be found under each subsection of the "Applications and Research Progress in Ruminants" section.
Comments 9: This introduction paragraph on conservation is missing in section 4, “Conservation of Genetic Resources”.
Response 9: Thank you for pointing this out. We also agree with this opinion. Therefore, we consulted the references and added the introductory content of the protection section, introducing the background knowledge of ruminants, pointing out that the genetic resources of ruminants are in an urgent state, and that the resequencing technology can be used to protect ruminants. This modification can be found on page 14, line 7 of the article.
Comments 10:All figures need to be self-explanatory. For example, please give a brief description of the main points you want to highlight in Figure 2. Did you develop Figure 2 by yourself? If you have used the data or resources from other articles, please add those references in the Figure legend.
Response 10: Thank you for pointing this out. We also agree with this opinion. Thank you for pointing this out. We also agree with this opinion. Note: This is a summary chart of the application of whole-genome resequencing technology in the aspects of population genetic structure, genetic diversity, economic traits, and adaptive traits of ruminants.
Comments 11: In each section, please explain what the main highlights generated from WGS are in these domains. For example, specify at least one example for each section. In section 3.1 Population Genetic Structure Analysis, please discuss the key points we got from WGS and how WGRS added to our current knowledge. Then, explain how resequencing helped in improving our understanding of the whole domain.
Response 11: Thank you for pointing this out. We also agree with this opinion. We pointed out that whole-genome sequencing can obtain the reference genome of a species, and whole-genome resequencing (WGRS) can help us understand the genetic diversity and population genetic structure of a species population. This modification can be found on line 39 of page 6 of the article.
Comments 12: I would add future perspective as the fifth section and add conclusion as the sixth section. Please limit the length of the conclusion section. Please summarise the key message of the article (take-home messages) in the conclusion section, avoiding references and further explanations.
Response 12: Thank you for pointing this out. We also agree with this opinion. We added "Future Perspectives" as the fifth section of the article, in which we explored the possible future development directions and potential research points in the research field, presenting the prospects for follow-up research to readers. Meanwhile, we added "Conclusion" as the sixth section. These modifications can be found on page 19 of the article.
Round 2
Reviewer 1 Report
Comments and Suggestions for Authors
"grey" is incorrectly spelt on page 7
Author Response
Comments1: "grey" is incorrectly spelt on page 7
Response 1: Thank you for pointing this out. Since this section does not fall within the scope of Asia and is not relevant to the current article's topic, we have decided to remove it. As a result, we will not be correcting the aforementioned spelling error.
Reviewer 2 Report
Comments and Suggestions for Authors
The authors have substantially modified the manuscript addressing all the comments.
Please see a few more minor corrections to be made before publication.
Title
Leveraging Whole-Genome Re sequencing to Uncover Genetic Diversity and Promote Conservation Strategies for Ruminants in Asia
Is the review pertaining to Asia only?
Table 2-Please change Breed into Species or animal since the column describes different animals such as sheep and cattle, not breed. I would advise to prepare the Table 2 as of Table 1. Table 1 seems more self-explanatory.
Figure 1. Please modify the text to more readable format. Please delete the note below Figure 1. Instead, I would add a short description of the figure. What do you like to communicate with this figure? I cannot see WGRS or WGS in the figure. I would narrate the transition from WGS to WGRS, which may reflect the topics discussed in the review. A summary chart as you mentioned is intended to highlight the summary of the article.
Alternatively, if you keep this figure, I advise to use it as a graphical abstract.
Author Response
Comments 1:“Leveraging Whole-Genome Re sequencing to Uncover Genetic Diversity and Promote Conservation Strategies for Ruminants in Asia” Is the review pertaining to Asia only?
Response 1: Thank you for pointing this out. We agree with this comment. Therefore, we have checked the content of the article and deleted or replaced the parts that are not related to Asia.
Comments 2:Table 2-Please change Breed into Species or animal since the column describes different animals such as sheep and cattle, not breed. I would advise to prepare the Table 2 as of Table 1. Table 1 seems more self-explanatory.
Response 2:Thank you very much for your suggestion, and we also agree with this point. I will reorganize Table 2 in the format of Table 1.
Comments 3:
Figure 1. Please modify the text to more readable format. Please delete the note below Figure 1. Instead, I would add a short description of the figure. What do you like to communicate with this figure? I cannot see WGRS or WGS in the figure. I would narrate the transition from WGS to WGRS, which may reflect the topics discussed in the review. A summary chart as you mentioned is intended to highlight the summary of the article. Alternatively, if you keep this figure, I advise to use it as a graphical abstract.
Response 3:Thank you for pointing this out. We agree with this comment. After careful consideration, we have concluded that this image does not align well with the content of the article, and we have decided to remove it.